# The Distinct Innate Immune Response of Warm Ischemic Injured Livers during Continuous Normothermic Machine Perfusion

**DOI:** 10.3390/ijms241612831

**Published:** 2023-08-16

**Authors:** Joris Blondeel, Nicholas Gilbo, Veerle Heedfeld, Tine Wylin, Louis Libbrecht, Ina Jochmans, Jacques Pirenne, Hannelie Korf, Diethard Monbaliu

**Affiliations:** 1Department of Abdominal Transplant Surgery and Transplant Coordination, University Hospitals Leuven, 3000 Leuven, Belgium; joris.blondeel@kuleuven.be (J.B.); nicholas.gilbo@chuliege.be (N.G.); veerle.heedfeld@kuleuven.be (V.H.); tine.wylin@kuleuven.be (T.W.); ina.jochmans@uzleuven.be (I.J.); jacques.pirenne@uzleuven.be (J.P.); 2Laboratory of Abdominal Transplantation, Department of Microbiology, Immunology and Transplantation, KU Leuven, 3000 Leuven, Belgium; 3Department of Pathology, AZ Groeninge, 8500 Kortrijk, Belgium; louis.libbrecht@kuleuven.be; 4Laboratory of Hepatology, CHROMETA Department, KU Leuven, 3000 Leuven, Belgium; hannelie.korf@kuleuven.be

**Keywords:** liver transplantation, ischemia-reperfusion injury, normothermic machine perfusion, preservation, cytokines

## Abstract

Although normothermic machine perfusion (NMP) provides superior preservation of liver grafts compared to static cold storage and allows for viability testing of high-risk grafts, its effect on the liver immune compartment remains unclear. We investigated the innate immune response during 6 h of continuous NMP (cNMP) of livers that were directly procured (DP, n = 5) or procured after 60 min warm ischemia (WI, n = 5), followed by 12 h of whole blood (WB) reperfusion. WI livers showed elevated transaminase levels during cNMP but not after WB reperfusion. Perfusate concentrations of TNF-α were lower in WI livers during cNMP and WB reperfusion, whereas IL-8 concentrations did not differ significantly. TGF-β concentrations were higher in WI livers during NMP but not after WB reperfusion, whereas IL-10 concentrations were similar. Endoplasmic stress and apoptotic signaling were increased in WI livers during cNMP but not after WB reperfusion. Additionally, neutrophil mobilization increased to a significantly lesser extent in WI livers at the end of NMP. In conclusion, WI livers exhibit a distinct innate immune response during cNMP compared to DP livers. The cytokine profile shifted towards an anti-inflammatory phenotype during cNMP and WB reperfusion, and pro-apoptotic signaling was stronger during cNMP. During WB reperfusion, livers exhibited a blunted cytokine release, regardless of ischemic damage, supporting the potential reconditioning effect of cNMP.

## 1. Introduction

The success of liver transplantation (LT) in the treatment of end-stage liver failure has resulted in an important imbalance between the numbers of patients on the waiting list and donor organ availability [1]. To overcome this imbalance, transplant clinicians are accepting more high-risk liver grafts from extended criteria donors, presenting characteristics such as advanced age or hepatic steatosis, as well as donation after circulatory death (DCD) donors, which are exposed to a period of warm ischemia (WI) prior to procurement, aggravating subsequent ischemia-reperfusion injury (IRI) [2,3]. As a result, these high-risk DCD grafts display higher rates of post-transplant complications such as early allograft dysfunction and ischemic cholangiopathy, which in turn result in lower graft survival [4,5].

Static cold storage (SCS) is the gold standard preservation method, but it falls short in adequately protecting these DCD livers from the inevitable IRI. Normothermic machine perfusion (NMP) has been proposed as an alternative preservation strategy [6], additionally providing the opportunity for objective graft assessment during the preservation phase, and therefore decreasing the persistently high discard rates observed in DCD livers.

During NMP, the liver is connected ex situ to a perfusion device that recirculates an oxygenated and nutrient-enriched packed red blood cell perfusate at physiological body temperature. When used to replace preservation with SCS, NMP is applied throughout the entire preservation phase and is also referred to as continuous NMP (cNMP). Alternatively, NMP can be used as a means to assess graft quality after a period of SCS or hypothermic machine perfusion, in which case it is referred to as end-ischemic NMP (eNMP) [2]. A recent randomized trial reported a reduction in early graft dysfunction and ischemic cholangiopathy in DCD livers preserved with cNMP [7]. Additionally, several non-randomized trials have demonstrated the feasibility of selecting and successfully using high-risk grafts for transplantation based on perfusate and bile markers during eNMP [8,9,10].

To date, most mechanistic studies on NMP have focused on hepatocellular and cholangiocellular function and injury. In contrast, little is known about the effects of NMP on the immune compartment of the liver. Lee et al. used eNMP to investigate the immune profile of six discarded human livers and showed that dynamic changes in leukocyte frequencies occurred, with the mobilization of neutrophils and T cells, together with significant secretion of pro-inflammatory cytokines [11]. Equally, Ohman et al. observed the activation of innate immune pathways during eNMP [12]. However, neither study compared profiles of DCD and donation after brain death (DBD) livers. Jassem et al. observed that cNMP, relative to SCS, was associated with a significant downregulation of pro-inflammatory genes, reduction in neutrophil infiltration, and upregulation of tissue regeneration gene expression in human livers [13]. However, liver tissue samples were taken exclusively at the end of preservation, and it is unknown whether NMP prevented rather than repaired inflammation.

The aim of the current study was to provide insights into the innate immune response during cNMP and subsequent whole blood (WB) reperfusion as a surrogate for transplantation, as well as how this is affected by prior exposure to WI. To this end, porcine livers were either procured directly (directly procured, DP group) or after 60 min of WI (WI group). All livers underwent 6 h of NMP followed by 12 h of allogenic WB reperfusion, and a comparative assessment of hepatocellular injury and function, cytokine secretion, endoplasmic reticulum (ER) stress, and leukocyte mobilization was performed.

## 2. Results

### 2.1. Warm Ischemic Livers Display More Severe Injury

Animal and perfusion characteristics were comparable between both groups (Table 1). Livers in the WI group suffered more hepatocellular damage as reflected by significantly higher aspartate transaminase (AST) release during NMP (AUC 2121 [1482–2528]) compared to the DP group (209 [125–387], *p* = 0.008). This difference was no longer significant during WB reperfusion (2372 [1791–4387] vs. DP 1378 [784–2365], *p* = 0.1) (Figure 1A,C). Tissue samples taken after one hour of NMP showed significantly more injury in the WI group (Suzuki score 3 [2–5] vs. DP 1 [0–2.5], *p* = 0.03) (Figure 1E,F). In contrast, hepatocellular function as assessed by lactate clearance was similar in both groups during cNMP (WI, 27 [10–52] vs. DP 10 [8–40], *p* = 0.3) and during WB reperfusion (WI 27.5 [18.5–54.6] vs. DP 30.7 [23.6–59.2], *p* = 0.7) (Figure 1B,D). Equally, there were no differences in intrahepatic vascular resistance between groups during NMP and WB reperfusion (Figure 1G,H).

### 2.2. Warm Ischemic Injured Livers Exhibit an Anti-Inflammatory Phenotype during Continuous NMP

The overall cytokine release during NMP and subsequent WB reperfusion was assessed by calculating the AUC of the cytokine perfusate concentration corrected for liver weight (Figure 2A,B). In both groups, Tumor Necrosis Factor-α (TNF-α) perfusate concentrations increased after one hour and peaked at three hours NMP; however, concentrations of TNF-α at three hours were significantly lower in WI (20.2 (10.6–58.9) pg/mL/g) than in DP (374.7 (256.1–482.8) pg/mL/g, *p* = 0.004) Figure 2C). Overall, in WI livers, there was a significant reduction in the production of the pro-inflammatory cytokine TNF-α both during NMP (AUC 7366 [4030–19,448] vs. DP 127,661 [91,903–168,234], *p* = 0.008) and WB reperfusion (AUC 1445 [843–7317] vs. 30,490 [22,431–36,661], *p* = 0.008). The same was observed for Interleukin-8 (IL-8) (NMP: WI AUC 50,204 [18,936–60,213] vs. DP 84,469 [50,356–143,392], *p* = 0.1; WB reperfusion: WI AUC 145,941 [57,862–197,331] vs. DP 210,450 [184,663–290,442], *p* = 0.1). Notably, IL-8 concentrations were only increased after three hours of NMP and peaked at six hours in both groups, with again significantly lower levels during NMP of WI (49.1 (26.9–108.7) pg/mL/g) than DP (205.6 (106.3–349.4) pg/mL/g, *p* = 0.047) at three hours. Moreover, we observed a distinct IL-8 dynamic after WB reperfusion compared to NMP in both groups (Figure 2D). Perfusate concentrations of Transforming Growth Factor (TGF)-β slowly increased throughout NMP and continued increasing after WB reperfusion (Figure 2E). In contrast to the pro-inflammatory cytokines, TGF-β perfusate concentrations were higher in WI livers primarily during NMP (NMP: AUC 1410 [1188–2070] vs. DP 641 [576–890], *p* = 0.02; WB reperfusion: AUC 6299 [5545–7304] vs. DP 3523 [4364–6863], *p* = 0.3). Perfusate concentrations of Interleukin (IL)-10 increased at three hours and peaked at six hours NMP, but overall, they were very low in both groups, and no significant difference could be observed during NMP (WI AUC 20.5 [6.4–123] vs. DP 117 [41–304], *p* = 0.2) or WB reperfusion (WI AUC 22.5 [16.7–93.1] vs. DP 25.9 [18.8–63.3], *p* > 0.99) (Figure 2F). To explore the balance between pro-inflammatory and anti-inflammatory signaling, we calculated the ratio of perfusate concentrations of anti-inflammatory (TGF-β and IL-10) cytokines and pro-inflammatory (TNF-α and IL-8) cytokines (Figure 3). The overall ratio between TGF-β and TNF-α was higher in the WI group compared to the DP group both during NMP (WI AUC 98.2 (39–114) vs. DP 13.9 (8.6–20), *p* = 0.06) and throughout WB reperfusion (WI AUC 149 (67.9–272) vs. DP 6.5 (3.1–8.4), *p* = 0.008). Equally, the TGF-β/IL-8 ratio was higher in the WI group during NMP (WI AUC 15.5 (11.8–22.6) vs. DP 3 (2.4–4.3), *p* = 0.016) and WB reperfusion (WI AUC 4.6 (2.8–12.7) vs. DP 0.8 (0.3–1), *p* = 0.057), whereas the IL-10/TNF-α was higher only during WB reperfusion (WI AUC 0.4 (0.3–0.8) vs. DP 0.03 (0.02–0.1), *p* = 0.008). No difference was observed between IL-10/IL-8 ratios.

### 2.3. Endoplasmic Reticulum-Specific Stress Signaling Is Stronger in WI Livers during NMP and Persists during WB Reperfusion

To investigate the effect of WI on unfolded protein response (UPR), which is activated in IRI to restore protein homeostasis and prevent cell death caused by the accumulation of misfolded proteins, we assessed the expression of two mRNAs known to be specifically induced upon endoplasmic reticulum (ER) stress: glucose-regulating protein-78 (GRP78), which is an ER chaperone, and C/EBP homologous protein (CHOP), a transcription factor that plays an important role in the ER stress-induced apoptosis pathway. Additionally, expression of the apoptosis-related downstream effectors B-cell lymphoma 2 (Bcl2) and Bcl2-associated X protein (Bax) were also measured. The expression of GRP78 was significantly increased in WI compared to DP livers, both during NMP (WI AUC 10.6 [8.1–12.6] vs. DP 2.8 [1.9–3.5], *p* = 0.008) and WB reperfusion (WI AUC 33.2 [16.8–60.7] vs. DP 7.9 [5–15.6], *p* = 0.03) (Figure 4B). CHOP expression was similarly increased in WI livers during NMP (WI AUC 59.2 [49–188.2] vs. DP 8.5 [6/9–16], *p* = 0.008) and WB reperfusion (WI AUC 81.7 [13–110.4] vs. DP 12.7 [6.7–23.2], *p* = 0.15) (Figure 4C). There was no difference in expression of the anti-apoptotic Bcl2 in WI livers, whereas expression of the pro-apoptotic Bax was increased in WI livers during NMP (WI AUC 6.8 [4.7–8.3] vs. DP 2.8 [2.2–4.2], *p* = 0.008) but not during WB reperfusion (WI AUC 20.5 [13.7–23] vs. DP 14.3 [10.7–26.5], *p* = 0.84) (Figure 4D,E). To visualize and quantify apoptosis, we performed TUNEL staining on biopsies taken at baseline, 1 h of NMP, and 6 h of NMP. Although statistically non-significant, the percentage of TUNEL-positive cells during NMP was higher in the WI than in the DP group (Figure 4F,G).

### 2.4. A Distinct Pattern in Leukocyte Mobilization

White blood cell frequencies were quantified using perfusate smears taken at different timepoints throughout NMP. Cell counts during WB reperfusion could not be performed, because total cell recovery was too low to determine accurate frequencies. No differences were observed during the first hour of NMP; however, at three and six hours, there were notable differences (Figure 5). Neutrophil frequency increased during NMP, with a greater increase observed in DP livers, which was significant at six hours (50.2% [48.9–51.3%] vs. 32.5% [26.2–37.2%], *p* = 0.015). The proportion of lymphocytes decreased in both groups throughout NMP, but this decrease was significantly greater in DP livers compared to WI livers at three hours (10.3% [3.4–14.2%] vs. 51.4% [31.8–56.1%, *p* = 0.015). The proportion of mononuclear cells remained similar throughout the NMP period in both groups.

## 3. Discussion

This study shows that livers exposed to severe WI damage display a distinct innate immune response during cNMP compared to livers that were directly procured, with a shift towards an anti-inflammatory cytokine profile and altered leukocyte mobilization. Despite suffering significantly more hepatocellular damage, as reflected by increased AST release, perfusate concentrations of pro-inflammatory cytokines TNF-α and IL-8 were significantly lower in WI livers during NMP. In contrast, IL-10 release was similar and TGF-β release was higher in WI livers, resulting in significantly greater TGF-β/TNF-α and TGF-β/IL-8 ratios. Moreover, proportionally fewer neutrophils were mobilized into the perfusate in WI livers. These findings imply that ischemic injury alters the liver’s biological response during NMP, suggesting different preservation modalities might suit livers of different quality.

DCD livers inevitably suffer a period of WI prior to the in situ cold flush, which aggravates IRI and the associated release of pro-inflammatory cytokines [14]. However, studies on the innate immune response after transplantation of WI-damaged livers are scarce. In a mouse model of DCD liver transplantation, Liu et al. showed that increasing durations of WI were associated with higher TNF-α tissue expression compared to livers that did not suffer WI [15]. In contrast to Liu et al., in a rat model, Saat et al. found that DBD livers exhibited increased TNF-α expression after procurement and during cold ischemia compared to DCD livers [16], which may be due to the cytokine storm associated with brain death [17]. Even though NMP is generally assumed to provide a near-physiological environment that mimics in vivo conditions, our study highlights that significant deviations may occur during NMP in response to WI compared to in vivo, since WI livers showed a reduced pro-inflammatory response compared to DP livers despite more severe hepatocellular damage (reflected by higher AST release and a higher Suzuki score). Although no brain death was induced in our DP group, TNF-α release was considerably higher in DP livers than in WI grafts, potentially implying a more pronounced protective effect of cNMP on DCD livers compared to DBD grafts. Indeed, in their randomized trial on NMP, Nasralla et al. showed that the relative reduction in post-transplant peak AST compared to cold storage was greater in DCD livers than DBD livers [18]. Cells that undergo WI injury and ultimately die may release an excessive amount of cytokines [19,20]. Nonetheless, the absence of necrosis on H&E staining, together with the observed rise in pro-apoptotic ER stress signaling in WI livers, suggests that cells subjected to WI are primed to undergo apoptosis during NMP. Apoptosis does not lead to the abundant release of pro-inflammatory cytokines typically associated with cell lysis [19], which could partially explain the reduced perfusate concentration of pro-inflammatory cytokines observed in WI livers. Of note, Hautz et al. compared cytokine levels in human DCD and DBD livers during end-ischemic NMP and did not identify significant differences, suggesting that the timing of NMP may considerably impact cytokine release, and cNMP provides a more favorable response compared to end-ischemic NMP [21]. Raigani et al. showed that inhibiting apoptosis in DCD livers during eNMP mitigated the innate immune response; whether this would also be beneficial during cNMP should be the subject of future investigations [22].

Along with a decreased pro-inflammatory response, WI livers displayed a higher release of TGF-β and, consequently, a shift in the balance towards anti-inflammatory signals. Jassem et al. showed that NMP of human livers upregulated activation of regenerative pathways [13]. TGF-β is known to participate in regenerative processes [23]; however, 6 h of NMP and 12 h of WB reperfusion in this study is likely too short to detect regeneration. An explanation for the more anti-inflammatory cytokine profile in WI livers might be found in reports from single-cell RNA sequencing on donor livers. Macparland et al. identified two distinct macrophage populations in the liver lobule [24], a pro-inflammatory and an anti-inflammatory phenotype. Cells of the anti-inflammatory macrophage population are mainly concentrated in periportal areas, making them the least susceptible to ischemia and potentially better preserved during cNMP. Perfusate cell counts did not reveal significant differences in mononuclear cell proportions; however, these analyses were not specific for macrophages.

Additionally, we investigated immune cell mobilization during NMP and observed a relative increase in neutrophils in the perfusate at 3 and 6 h. Interestingly, this increase was significantly lower in WI ischemic livers at the end of NMP. In a recent study, Hautz et al. described immune cell dynamics during NMP using single-cell RNA sequencing and flow cytometry [21]. They found neutrophils to be the predominant resident immune cells in the liver. In line with our findings, they observed a decrease in proportion of tissue neutrophils during NMP, which corresponded with an increase in perfusate neutrophils, suggesting significant neutrophil mobilization. Of note, tissue neutrophils were measured in 8 livers, 7 of which were DBD livers, whereas perfusate neutrophils were assessed in 34 livers, 25 of which were DBD livers. Our study suggests that significant differences occur depending on the nature of the ischemic damage. As neutrophils are involved in switching macrophages to an anti-inflammatory phenotype, this may also explain the observed increased TGF-β concentrations in WI livers [25]. As activated neutrophils can both contribute to tissue injury by releasing neutrophil extracellular traps and producing reactive oxygen species, as well as being involved in tissue repair by inhibiting T-cell cytotoxicity and removing cell debris [25], they may acquire a dual role depending on the nature of IRI.

During the WB reperfusion that followed the 6 h cNMP, the cytokine release pattern did not show the typical profile seen after hepatic IRI, regardless of exposure to WI. Whereas TNF-α typically reaches an early peak within 6 h after reperfusion, in our study, TNF-α concentrations remained relatively stable throughout the entire 12 h WB reperfusion phase. As TNF-α plays a critical role in propagating the pro-inflammatory responses, a blunted TNF-α response may protect the graft from IRI by creating a “less hostile” microenvironment. This is consistent with our observation that, at the end of WB reperfusion, there were no differences in AST release and histological injury in WI livers compared to controls despite the more severe injury at baseline. Our group has previously shown that porcine transplantation of livers that were exposed to 60 min WI and preserved with SCS inevitably led to primary non-function and recipient death [26]. In contrast, Schön et al. and Linares-Cervantes et al. were able to successfully transplant porcine livers exposed to up to 70 min of WI and preserved by cNMP [27,28], suggesting a potential reconditioning effect of NMP.

Our porcine experimental model inevitably presents with some limitations. First, one may argue that WB reperfusion with allogenic WB recruits alloreactive immune activity that should also be factored into the equation. However, early IRI mainly results from inflammatory responses generated by innate immune cells, whereas the adaptive immune response involved in alloreactive reactivity comes into play only at later stages, and hyperacute rejection in non-sensitized animals is very rare [29,30]. Moreover, in clinical LT, immunosuppressive therapy is not initiated earlier than 6 h after reperfusion of the graft in our center. Second, in vivo transplantation may be considered superior to WB reperfusion as a model to assess the potential reconditioning effect of NMP on WI-damaged livers. In large animal transplant models, it is difficult to account for the various biological confounders such as hydration, nutrition, and biological availability of medication, as well as wash-out of biomarkers during organ perfusion; however, the WB reperfusion setting offers a more standardized alternative that is suitable for mechanistic investigations such as the current study. Third, porcine models do present methodological limitations such as the limited availability of porcine ELISA and immunohistochemistry reagents. Nonetheless, this did not prevent us from adequately investigating the key cytokines and cell populations in hepatic in IRI. Additionally, one may argue that 60 min of WI is too aggressive and results in too much injury. However, in current clinical practice, liver grafts with WI times up to 30 min are regularly transplanted with acceptable results. Expansion of the donor pool can only be realized by exploring the potential and the limits of the reconditioning capacity of cNMP. Lastly, perfusate concentrations of different cytokines represent the end product of a complex cell signaling cascade and might not capture the full complexity of the innate inflammatory response. Future studies should investigate the potential association between the innate inflammatory response during cNMP of WI-damaged livers and a potential increase in regenerative signaling, as well as perform assays examining upstream signaling.

In conclusion, this study shows that livers exposed to WI exhibit a distinct innate immune response during cNMP compared to livers that are procured without WI damage. The cytokine profile of WI-damaged livers shifted towards a more anti-inflammatory phenotype during NMP, which persisted after WB reperfusion, whereas neutrophil mobilization during NMP was decreased in WI livers. Moreover, following six hours of NMP, a blunted cytokine release was observed during WB reperfusion, potentially indicating a protective effect against IRI. Whether this also improves long-term graft outcome should be the subject of future investigations. Overall, these findings imply that liver grafts respond differently to cNMP when suffering WI injury, potentially suggesting a greater protective effect on DCD livers, and therefore support the evolution towards more organ-tailored preservation, whereby different preservation modalities might suit organs of different quality.

## 4. Materials and Methods

### 4.1. Experimental Design and Endpoints

Porcine livers were either exposed to 60 min of WI (WI group, n = 5) before procurement or were directly procured (DP group, n = 5), followed by NMP for six hours. After six hours, the perfusate was changed to allogenic WB from a donor pig, and the liver was reperfused for 12 h to mimic transplantation (Figure 6). The allocation to the experimental group was randomized. To assess hepatocellular injury, perfusate levels of AST and histological injury were measured. Lactate clearance was measured as indication for hepatocellular function. TNF-α, IL-8, IL-10, and TGF-β were measured sequentially during NMP and WB reperfusion to assess inflammatory response. On liver biopsies taken at baseline, one hour, three hours, and six hours during NMP, GRP78, CHOP, Bcl-2, and Bax expression were measured to reflect endoplasmic reticulum (ER) stress and apoptotic signaling. Furthermore, immunohistochemistry was performed to assess apoptosis (TUNEL staining). Lastly, leukocyte mobilization was assessed through cell counts on perfusate smears. Perfusate samples were taken at baseline (before the liver was connected to the perfusion circuit), 15 min, 30 min, one hour, three hours, and six hours during NMP. During WB reperfusion, perfusate and tissue samples were taken at the same time point as during NMP, with an additional sampling at 12 h.

### 4.2. Animals

Male TOPIGS TN70 pigs (aged 3 months, Nijmegen, The Netherlands) with a body weight of approximately 30 kg and liver weight of 600–700 g were used for this study protocol. Animals were kept in the animal facility of the University Hospitals of Leuven under a 12 h day/night rhythm in single pens with free access to food (MPig-H; ssniff, Soest, Germany) and tap water, with visual, olfactory, and auditory contact between them. The animals were kept for at least two days prior to the surgery to get accustomed to their surroundings. Animals were euthanized by exsanguination under general anaesthesia during liver procurement or WB collection. The KU Leuven Animal Care Committee approved the study, and the experiments were conducted in line with European guidelines [31]. Results are reported following the ARRIVE guidelines [32].

### 4.3. Surgery

The surgical model has been reported in detail elsewhere [33,34]. Briefly, the inferior vena cava and abdominal aorta were dissected free and prepared for cannulation. The bile duct, portal vein, and hepatic artery were dissected free. After administration of 500 U/kg of heparin (Leo Pharma, Ballerup, Denmark), the liver was flushed with 4000 mL of cold (2–4 °C) Institute George Lopez 1 preservation solution (Institute George Lopez, Lissieu, France) via the aorta (2000 mL) and the portal vein (2000 mL). Blood was collected during liver flush-out and washed with a cell-saver (Electa; Sorin Group, Mirandola, Italy) to obtain concentrated red blood cells. After hepatectomy, the liver was cold preserved during back-table preparation until perfusion was started. The inferior vena cava, portal vein, and hepatic artery were cannulated, and the supra-hepatic inferior vena cava was closed. The cystic duct was ligated, and the bile duct was cannulated for bile collection during perfusion. Allogenic WB was obtained from a donor pig (n = 5) via direct canulation of the aorta and collected and stored in clinical-grade blood storage bags.

### 4.4. Normothermic Machine Perfusion

Porcine NMP experiments were performed as previously described [33]. Briefly, the portal vein was perfused by gravity and the hepatic artery by a centrifugal pump delivering a pressure-controlled non-pulsatile flow (Medtronic, Dublin, Ireland). The outflow was collected via the IVC cannula and recirculated by the centrifugal pump (Figure 7). The perfusate was oxygenated to reach a perfusate oxygen tension (pO_2_) of 70–100 mmHg and warmed up to 38 °C. The circuit was primed with 500 mL of Gelufosine (Braun, Melsungen, Germany).

During NMP, the volume of washed red blood cells to be added in order to reach a hematocrit of 30% was calculated with the formula: [(weight of the liver + priming volume) × desired hematocrit]/hematocrit after washing. The same formula was used for the allogenic WB reperfusion. The perfusate was supplemented with heparin, antibiotics, calcium gluconate, and bicarbonate as previously reported. In line with current clinical practice of liver NMP preservation, perfusion was started after a period of SCS of circa 90 min, which corresponds to the time needed for cold flush in the donor, hepatectomy, and back-table preparation. Vasodilators, insulin, heparin, and sodium taurocholate were continuously infused during NMP, as previously described.

### 4.5. Assays

#### 4.5.1. Intrahepatic Vascular Resistance

During perfusion, portal vein and hepatic artery flows (BioPro TT, flowtrack plus, Em-Tec, Finning, Germany) and pressures (Stöckert-Shiley, Sorin group, Munich, Germany) were continuously monitored, and pointwise data were recorded at 15 min, 30 min, one hour, three hours, and six hours during NMP (also at 12 h during WB reperfusion). Intrahepatic vascular resistance was calculated as the pressure (mmHg)/flow (mL/min) ratio.

#### 4.5.2. Hepatocellular Damage and Function

Perfusate samples were collected in EDTA tubes, centrifuged at 2876× *g* at 4 °C, aliquoted in 1 mL microtubes, snap frozen, and stored at −80 °C. AST was measured according to the International Federation of Clinical Chemistry method (detection limit 4 U/L; Hitachi/Roche Modular P Chemistry Analyzer; Roche Belgium, Vilvoorde, Belgium). Lactate perfusate concentration was measured with a point-of-care instrument (ABL 800-flex; Radiometer, Zoetermeer, The Netherlands).

#### 4.5.3. Histology

Tissue samples were stored in 4% formalin, paraffin-embedded, and stained with hematoxylin-eosin (H&E). Histological injury was scored semi-quantitively with the SUZUKI score [35] by an experienced pathologist (LL) who was blinded for the allocation to the experimental groups.

#### 4.5.4. Cytokine Levels

Enzyme-linked immunosorbent assays (ELISA) were used to determine perfusate concentrations of TNF-α, IL-8, IL-10, and TGF-β (Biotechne, R&D systems Inc., Minneapolis, MN, USA) following manufacturer instructions.

#### 4.5.5. ER Stress

Total RNA from frozen liver sections stored in Trizol (life technologies, Thermo Fisher Scientific, Carlsbad, CA 92008 USA) was isolated using miRNeasy Mini kit (Qiagen, 5912 PL Venlo, The Netherlands). The resultant RNA was quantified using a NanoDrop one (Thermo Fisher Scientific, Waltham, MA, USA). Subsequently, equal amounts of RNA were converted to complementary DNA using MMLV (Invitrogen, Thermo Fisher Scientific, Carlsbad, CA 92008 USA). Gene expression assays were performed using the following primers: CHOP (Ss03821509_s1), GRP78 (Ss03374255_m1), Bcl-2 (Ss03375167_s1), and Bax (Ss03375842_u1) (Applied Biosystems, Thermo Fisher Scientific, Waltham, MA, USA). Amounts of specific RNA were normalized to established housekeeping genes (β-actin) (Ss03376563_uH) (Applied Biosystems, Thermo Fisher Scientific, Waltham, MA, USA), which remained constant in the experimental groups.

#### 4.5.6. Immunohistochemistry

Apoptosis was quantified by terminal deoxynucleotidyltransferase dUTP nick-end labeling (TUNEL assay) using the In Situ Cell Death Detection Kit, TMR red (Merck (12,156,792,910, Darmstadt, Germany)) on formalin-fixed, paraffin-embedded liver tissue samples. Five µm thin sections were prepared on Superfrost Plus Gold adhesion microscope slides, deparaffinized (2× 5 min in xylene), rehydrated (subsequently, 3 min in 100%, 96%, 70%, and 50% absolute ethanol), and rinsed in phosphate-buffered saline (PBS) (3× 5 min) at room temperature. After a Proteinase K treatment (Merck, Darmstadt, Germany) (1.24568.0100) (20 µg/mL in PBS for 15 min at 37 °C), slides were rinsed (4× 5 min in PBS at room temperature) and incubated with the TUNEL reaction mixture for 1 h at 37 °C in a humidified chamber. Afterwards, the slides were rinsed with PBS (4× 5 min at room temperature), and nuclei were stained with DAPI (1/2000 diluted in PBS) (ThermoScientific (62,248), Waltham, MA, USA). After a final wash, a drop of Slow Fade Diamond antifade (Fisher Scientific (15,441,244), Waltham, MA, USA) was applied. Staining was visualized using a fluorescence microscope (TUNEL (Ex 540/Det 580); DAPI (Ex372/Det456)).

Positive and negative deoxynucleotidyl transferase dUTP nick-end labeling cells were counted by an assessor (TW) who was blinded for the group allocation on 4 to 8 complete sections per sample at 20× magnification, then the apoptosis index was calculated as the ratio between the number of positive cells and the number of total cells.

### 4.6. Statistical Analyses

As this was a preliminary explorative study, a sample size calculation was not possible. The sample size in each group was chosen to account for biological variation between animals. Data are summarized as median (interquartile range). Data were plotted in relation to perfusion time and single timepoints. To assess the overall inflammatory response, the AUCs for cytokine, AST, and lactate perfusate concentrations were calculated and compared between groups with a Mann–Whitney U test. A similar approach was used for ER stress-related markers in tissue samples. Complete case analysis was used for dealing with missing data in AUC calculation. Concentrations below the inferior detection limit of the used assays were rounded to said value for the calculation of the AUCs. To explore the balance between anti- and pro-inflammatory signals, ratios of perfusate concentrations of IL-10 and TGF-β to that of TNF-α and IL-8 were calculated at every time point and compared between groups. Correction for multiple testing was performed with the Sidak method when appropriate. A *p*-value of <0.05 in a two-sided test was considered significant. Statistical analyses were performed with GraphPad Prism (GraphPad Software, San Diego, CA, USA, version 9).

## Figures and Tables

**Figure 1 ijms-24-12831-f001:**
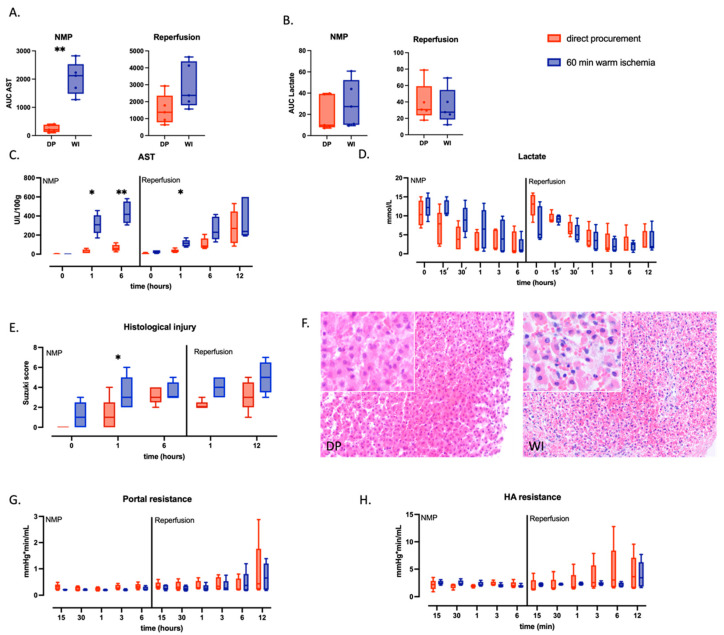
Hepatocellular injury and function, histological damage, and intrahepatic vascular resistance during NMP and reperfusion of directly procured livers (DP—red) or livers exposed to 60 min of WI prior to procurement (WI—blue). Boxes represent median and interquartile range; whiskers represent total range. (**A**,**B**) Area under the curve for AST (**A**) and lactate (**B**) during entire NMP and reperfusion. (**C**,**D**) Longitudinal evolution of perfusate concentrations of AST (**C**) and lactate (**D**). (**E**) Longitudinal evolution of histological injury as quantified by the Suzuki score. (**F**) Micrographs from representative H&E-stained sections pf a DP (left) and WI (right) liver, taken at the end of reperfusion: 200× magnification, insert at 400× magnification). (**G**,**H**) Longitudinal evolution of the portal resistance (**G**) and hepatic artery (HA) resistance (**H**). * *p* < 0.05 and ** *p* < 0.01 as tested by pairwise comparison using two-way ANOVA at the indicated timepoint; AUC, area under the curve; DP, direct procurement; WI, WI; AST, aspartate transaminase; NMP, normothermic machine perfusion.

**Figure 2 ijms-24-12831-f002:**
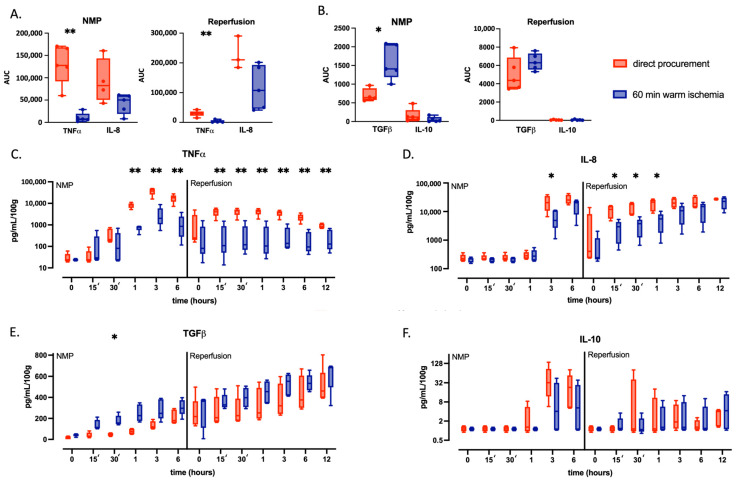
Perfusate concentrations, normalized for liver weight, during NMP and reperfusion, of directly procured livers (DP—red) and livers exposed to 60 min of WI prior to procurement (WI—blue). Boxes represent median and interquartile range; whiskers represent total range. (**A**,**B**) AUCs of all cytokines during NMP and reperfusion. (**C**–**F**). Longitudinal evolution of perfusate cytokine concentrations during NMP and reperfusion. * *p* < 0.05 and ** *p* < 0.01; AUC for pairwise comparisons with two-way ANOVA at the indicated timepoint. AUC, area under the curve; NMP, normothermic machine perfusion; TNF-α, Tumor Necrosis Factor-α; IL-8, interleukin-8; IL-10, interleukin-10; TGF-β, Transforming Growth Factor-β.

**Figure 3 ijms-24-12831-f003:**
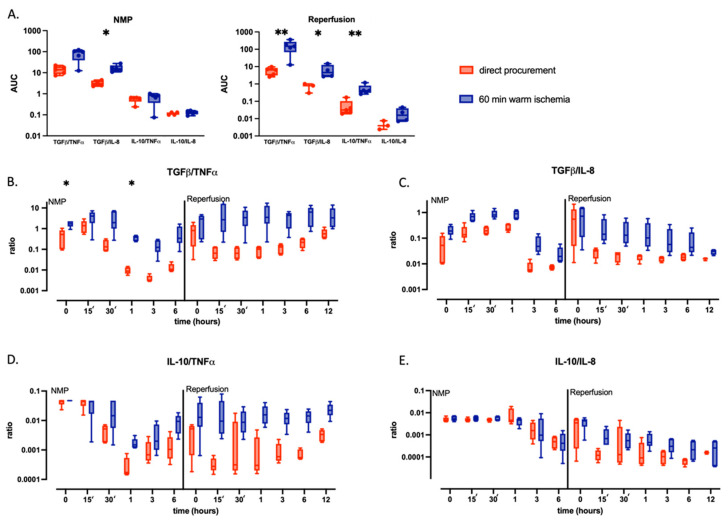
Ratios of perfusate concentrations of pro- (TNF-α and IL-8) and anti-inflammatory (TGF-β and IL-10) cytokines during NMP and reperfusion of directly procured livers (DP—red) and livers exposed to 60 min of WI prior to procurement (WI—blue). Boxes represent median and interquartile range; whiskers represent total range. The AUC (**A**) and longitudinal evolution (**B**–**E**) of ratios between anti- and pro-inflammatory cytokines are shown. * *p* < 0.05 and ** *p* < 0.01; AUC for pairwise comparison with two-way ANOVA at the indicated timepoint. AUC, area under the curve; NMP, normothermic machine perfusion; TNF-α, Tumor Necrosis Factor-α; IL-8, interleukin-8; IL-10, interleukin-10; TGF-β, Transforming Growth Factor-β.

**Figure 4 ijms-24-12831-f004:**
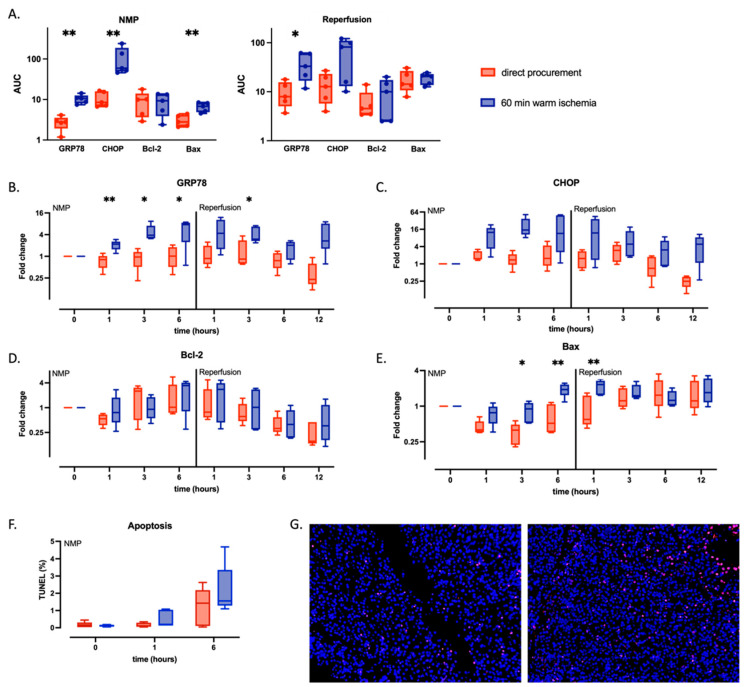
Tissue expression of ER stress-related transcription factors during NMP and WB reperfusion in directly procured livers (DP—red) or livers exposed to 60 min of WI prior to procurement (WI—blue). Boxes represent median and interquartile range; whiskers represent total range. Shown are the AUC’s of ER stress related gene expression (**A**), the longitudinal evolution of tissue gene expression of GRP78 (**B**) and CHOP (**C**) during NMP and reperfusion, longitudinal evolution of tissue gene expression of Bcl-2 (**D**) and Bax (**E**) during NMP and reperfusion, and longitudinal evolution of proportions of TUNEL-positive cells during NMP (**F**). Micrograph from a representative TUNEL-stained section (**G**) is shown at 20× magnification (DP liver on the left, WI liver on the right). * *p* < 0.05 and ** *p* < 0.01 for pairwise comparisons with two-way ANOVA at the indicated timepoint; NMP, normothermic machine perfusion; GRP78, glucose-regulated protein-78; CHOP, C/EBP homologous protein; Bcl-2, B-cell lymphoma 2; Bax, Bcl2-associated X protein.

**Figure 5 ijms-24-12831-f005:**
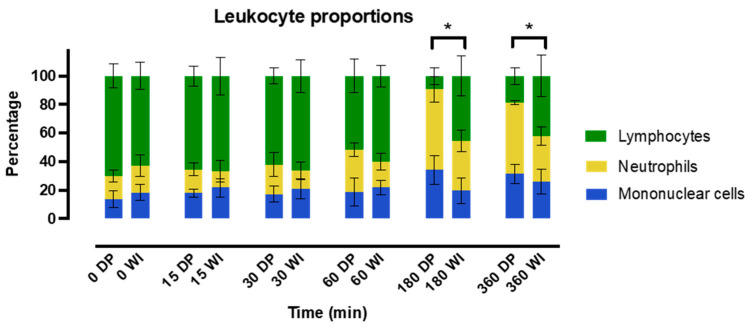
Shown are the leukocyte proportions in the perfusate as determined by cell counting at different timepoints during NMP in directly procured (DP) and 60 min warm ischemia (WI)-exposed livers. * *p* < 0.05, error bars represent interquartile range; DP, direct procured livers; WI, warm ischemic livers.

**Figure 6 ijms-24-12831-f006:**
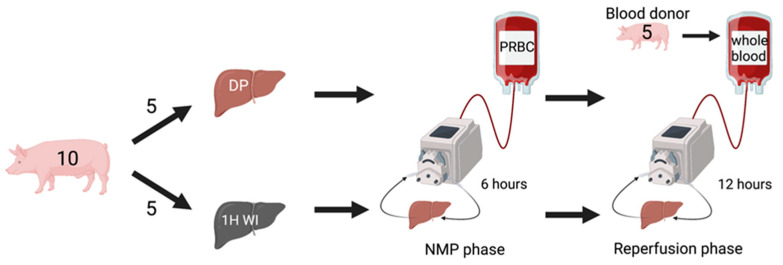
Experimental design. Pigs were randomized in two groups: in one group, the liver was exposed to 1 h of WI prior to cold flush and procurement (1H WI), whereas in the other group, the livers were procured without WI (directly procured, DP). After procurement and back table preparation, livers underwent 6 h of normothermic machine perfusion (NMP). After 6 h NMP, livers were removed from the machine and cold flushed; the packed red blood cell perfusate was changed to allogenic WB perfusate, and livers were reperfused for 12 h to mimic transplantation. DP, directly procured; WI, WI, NMP, normothermic machine perfusion; PRBC, packed red blood cells.

**Figure 7 ijms-24-12831-f007:**
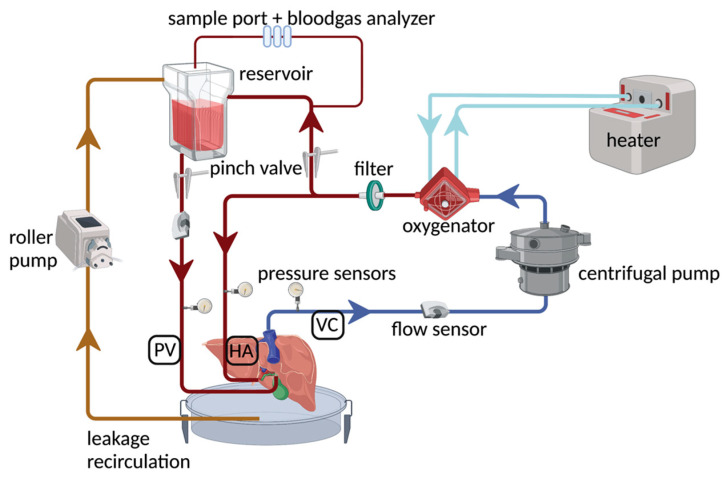
Schematic presentation of the perfusion circuit, as previously described [33]. Briefly, the liver is perfused in a closed circuit, with canulation of the portal vein, hepatic artery, and infrahepatic vena cava, and the suprahepatic vena cava is closed. Portal inflow is provided from a reservoir, where the pressure is determined by the hydrostatic column, whereas arterial inflow is directly provided by the centrifugal pump. An oxygenator is incorporated in the circuit, and a heater enables perfusion at 38 °C.

**Table 1 ijms-24-12831-t001:** Animal and perfusion characteristics.

	WI	DP	*p*-Value
Animal weight (kg)	30 (28.9–33.8)	30 (26.6–33.8)	0.81
Liver weight (g)	806 (713–858)	770 (648–919)	0.84
Cold ischemia time (min)	89 (79–111)	83 (77–100)	0.6
Hemoglobin (g/dL)			
Start NMP	10.0 (6.5–11.6)	9.3 (8.6–13.1)	0.69
End NMP	5.8 (4–7.8)	7 (6.8–7.7)	0.45
Start reperfusion	6.7 (6.3–7.1)	7.4 (7.1–8.3)	0.04
End reperfusion	5 (3.9–7)	3.5 (3.2–8.5)	>0.99
pH			
Start NMP	7.12 (6.85–7.31)	7.39 (7.29–7.49)	0.056
End NMP	7.42 (7.34–7.61)	7.34 (7.32–7.46)	0.55
End reperfusion	7.43 (7.24–7.48)	7.51 (7.42–7.58)	0.095
pO_2_ (mmHg)			
Start NMP	92 (79–154)	99 (74–103)	0.95
End NMP	88 (79–126)	95 (83–99)	>0.99
End reperfusion	110 (78–125)	86 (57–93)	0.15

NMP, normothermic machine perfusion; WI, WI; DP, direct procurement; pO_2_, partial tension of oxygen. Data are summarized with mean (±SD) and compared with Mann–Whitney U test.

## Data Availability

All data are available upon request from the corresponding author.

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
