# Peer review of "The Distinct Innate Immune Response of Warm Ischemic Injured Livers during Continuous Normothermic Machine Perfusion"

_ijms, 2023, doi:10.3390/ijms241612831_

Round 1

Reviewer 1 Report

This is a nice paper from the group in Leuven, which builds on their prior work studying the effects of NMP on livers. The authors compared 5 pig livers that underwent direct to NMP versus 60 minutes of WI followed by NMP, then whole blood reperfusion. In contrast to what would be assumed, the WI livers demonstrated more anti-inflammatory perfusate markers compared to the DP livers. WI livers however did demonstrate more ER stress signaling and pro-apoptotic signaling as would be predicted. The results are a little contradictory but make for a good story in the right context and the authors are commended on a well-controlled study and nicely written manuscript.

Major comments

- I wonder if the surprising anti-inflammatory findings in the WI group are due to the fact that the authors looked at the end-product of cell signaling. Release of cytokines into the serum represent the downstream effect of cell signaling and IRI. Perhaps if additional studies examining the upstream signaling were performed, we would potentially see more intra-cellular pro-inflammatory signaling at the mRNA level. This idea is supported by the results from the western blot analyses, which demonstrate cell signaling in response to inflammation and ischemia. Additional discussion in this area would be relevant. Lack of specific studies of mRNA-level analysis should be addressed.

- several key references are missing with respect to work done in the field of liver IRI/innate immunity/ER stress/apoptosis during NMP that predate this study and provide further context for discussion (PMID 35958587, 34730028)

- If possible, I would strongly recommend additional analysis be performed for apoptosis as TUNEL is tends to be nonspecific. ELISA or serum based assays of caspase activity are relatively easy to perform or tissue based analysis (Western blot or IHC for cleaved caspase-3)

- Another thought on the muted results between the two groups in certain analyses is that 1hr WI is not enough to see a significant difference. A high-performing team such as the Leuven group likely are able to get the pig livers on the perfusion device quite expeditiously. Adding additional cold storage time or multiple WI time periods in future experiments may better delineate the liver's response to IRI.

- I don't agree that whole blood reperfusion is a better clinical model of study compared to large animal transplantation. It's certainly easier to study and extract more data with WB reperfusion but transplantation after NMP should remain the gold standard for experimental surgery and this should be addressed in the limitations.

Minor comments

- Whole blood abbreviation is missing

- very minor grammatical errors ("raised" as opposed to "increased")

very good, minor edits required

Author Response

Dear Editors and reviewers,

We would sincerely like to express our gratitude for your decision to give us the chance to revise our manuscript based on the comments of the reviewers. We believe that addressing these comments has substantially improved our manuscript. Please find below a detailed outline of our answers to the editors and reviewers.

Sincerely,

On behalf of all co-authors,

Joris Blondeel

Reviewer 1:

This is a nice paper from the group in Leuven, which builds on their prior work studying the effects of NMP on livers. The authors compared 5 pig livers that underwent direct to NMP versus 60 minutes of WI followed by NMP, then whole blood reperfusion. In contrast to what would be assumed, the WI livers demonstrated more anti-inflammatory perfusate markers compared to the DP livers. WI livers however did demonstrate more ER stress signaling and pro-apoptotic signaling as would be predicted. The results are a little contradictory but make for a good story in the right context and the authors are commended on a well-controlled study and nicely written manuscript.

Dear reviewer,

Thank you for your feedback on our manuscript. We agree that these results are somewhat contra-intuitive, but therefore all the more interesting, in our opinion. We attempted to incorporate your comments as much as possible and hope that you are satisfied with the changes to the manuscript.

Major comments:

- I wonder if the surprising anti-inflammatory findings in the WI group are due to the fact that the authors looked at the end-product of cell signaling. Release of cytokines into the serum represent the downstream effect of cell signaling and IRI. Perhaps if additional studies examining the upstream signaling were performed, we would potentially see more intra-cellular pro-inflammatory signaling at the mRNA level. This idea is supported by the results from the western blot analyses, which demonstrate cell signaling in response to inflammation and ischemia. Additional discussion in this area would be relevant. Lack of specific studies of mRNA-level analysis should be addressed.

Thank you for this comment. Cytokine release into the perfusate is indeed the downstream result of several cell signaling pathways and by itself does not completely capture the complexity of the innate inflammatory response. However, we believe that measuring perfusate concentrations does hold significant value by itself. Nevertheless, we agree that more in depth research on intracellular pathways would be very interesting. We addressed this in the discussion (p10 line 323).

- several key references are missing with respect to work done in the field of liver IRI/innate immunity/ER stress/apoptosis during NMP that predate this study and provide further context for discussion (PMID 35958587, 34730028)

Thank you for this comment, we have incorporated these in the manuscript.

- If possible, I would strongly recommend additional analysis be performed for apoptosis as TUNEL is tends to be nonspecific. ELISA or serum based assays of caspase activity are relatively easy to perform or tissue based analysis (Western blot or IHC for cleaved caspase-3)

Thank you for this comment. We agree that TUNEL can be nonspecific, however, we do believe these results are in line with the observed stronger endoplasmic stress related pro-apoptotic signaling. While we would be glad to explore other assays, given the 5-days timeframe granted by the editors it is not possible to perform these additional analyses and include them in this resubmission.

- Another thought on the muted results between the two groups in certain analyses is that 1hr WI is not enough to see a significant difference. A high-performing team such as the Leuven group likely are able to get the pig livers on the perfusion device quite expeditiously. Adding additional cold storage time or multiple WI time periods in future experiments may better delineate the liver's response to IRI.

Thank you for this comment. Indeed, choosing the amount of damage in a study on ischemia-reperfusion injury is always somewhat arbitrary. First, we chose one hour of warm ischemia based on our previous experiences where in a pig transplant model, one hour of warm ischemia followed by transplantation invariably led to primary non function. Additionally, in previous NMP studies, we found that one hour of warm ischemia resulted significant differences in production of coagulation factors, as well as distinct content of extracellular vesicles isolated from the perfusate and bile (data not published). These findings, together with the significant difference in perfusate AST between groups, indicate that one hour of warm ischemia does provide adequate damage to observe relevant outcomes differences. Nonetheless, we agree that it would be interesting to investigate NMP of livers exposed to even longer periods of warm ischemia in future studies. Additionally, as we wanted to investigate continuous NMP, adding any cold storage time would not be desirable for this design. The innate immunity response during end ischemic NMP may be different from that of cNMP, therefore future studies comparing the inflammatory fingerprint of cNMP vs. eNMP of 60min WI livers may be highly informative, but the subject of an entire novel study.

- I don't agree that whole blood reperfusion is a better clinical model of study compared to large animal transplantation. It's certainly easier to study and extract more data with WB reperfusion but transplantation after NMP should remain the gold standard for experimental surgery and this should be addressed in the limitations.

Thank you for your comment. We agree that large animal transplant survival models are the gold standard for studying transplant function. Our argument for the advantage of whole-body reperfusion is in the context of mechanistic studies. In such studies, it is necessary to minimize all potential sources of inter-subject variability as much as possible, especially with large animals where sample sizes are small. We clarified this in the manuscript (p10 line 311).

Minor comments

- Whole blood abbreviation is missing

- very minor grammatical errors ("raised" as opposed to "increased")

Thank you for these remarks, we adjusted accordingly.

Reviewer 2 Report

Dear authors,

Your paper describes a very thorough analysis of the innate immune response of livers preserved with NMP. The lower production of pro-inflammatory cytokines described in the WI livers is an unexpected finding but would support the concept that the more injured livers benefit the most from cNMP.  I believe, this may be different for end-ischaemic NMP.

Minor comment:

-Page 10 line 305, it says that immunosuppressive therapy is not initiated earlier than 6h after reperfusion. I don't think this is fully correct as high dose steroids is routinely given during the anhepatic phase. Can you please clarify?

Author Response

Your paper describes a very thorough analysis of the innate immune response of livers preserved with NMP. The lower production of pro-inflammatory cytokines described in the WI livers is an unexpected finding but would support the concept that the more injured livers benefit the most from cNMP.  I believe, this may be different for end-ischaemic NMP.

Minor comment:

-Page 10 line 305, it says that immunosuppressive therapy is not initiated earlier than 6h after reperfusion. I don't think this is fully correct as high dose steroids is routinely given during the anhepatic phase. Can you please clarify?

Dear reviewer,

Thank you for these comments. Concerning the immunosuppressive therapy, in our center, we don’t generally administer steroids during the anhepatic phase. We nuanced in the manuscript that this might be specific to our center. (p10 line 305)